# Human Amniotic Suspension Allograft Improves Pain and Function in Knee Osteoarthritis: A Prospective Not Randomized Clinical Pilot Study

**DOI:** 10.3390/jcm11123295

**Published:** 2022-06-08

**Authors:** Simone Natali, Luca Farinelli, Daniele Screpis, Diletta Trojan, Giulia Montagner, Francesca Favaretto, Claudio Zorzi

**Affiliations:** 1Department of Orthopaedics IRCCS Ospedale Sacro Cuore Don Calabria, 37024 Negrar di Valpolicella, Italy; daniele.screpis@sacrocuore.it (D.S.); claudio.zorzi@sacrocuore.it (C.Z.); 2Clinical Ortopaedics, Department of Clinical and Molecular Sciences, Università Politecnica delle Marche, 60121 Ancona, Italy; farinelli.luca92@gmail.com; 3Fondazione Banca dei Tessuti di Treviso Onlus, 31100 Treviso, Italy; dtrojan@fbtv-treviso.org (D.T.); gmontagner@fbtv-treviso.org (G.M.); ffavaretto@fbtv-treviso.org (F.F.)

**Keywords:** osteoarthritis, knee, mesenchimal stem cells (MSC), human amniotic membrane, human amniotic allograft, orthobiologic, biological therapies, innovative therapies

## Abstract

Osteoarthritis (OA) is a chronic debilitating disorder causing pain and gradual degeneration of joints. Among various cell therapies, mesenchymal stem cell (MSC) therapy appears to provide encouraging results. Human amniotic suspension allografts (HASA) have anti-inflammatory and chondroregenerative potential and represent a promising treatment strategy. The purpose of the present study was to prospectively assess the safety, clinical effectiveness, and feasibility of intra-articular injections of human amniotic suspension allograft (HASA) in unilateral knee OA in order to assess the improvement of symptoms and delay the necessity for invasive surgical procedures. A total of 25 symptomatic patients, affected by knee OA were treated with 3 mL of HASA. Clinical evaluations before the treatment and after 3, 6, and 12 months were performed through International Knee Documentation Committee (IKDC) score and Visual Analogue Scale (VAS) scores. Adverse events were recorded. No severe complications were noted during the treatment and the follow-up period. A statistically significant improvement from basal evaluation to the 3-, 6-, and 12-month follow-up visits was observed. The present pilot study indicates that a single intra-articular injection of HASA seems safe and able to provide positive clinical outcomes, potentially offering a new minimally invasive therapeutic option for patients with knee OA.

## 1. Introduction

Osteoarthritis (OA) is characterized by cellular stress and extracellular matrix degradation within synovial joint. These processes are initiated by injury, followed by maladaptive repair responses, with activation of innate immunity pro-inflammatory pathways [1,2,3]. Currently, there are few effective treatments for OA, and none prevent disease progression. Non-operative treatments consist of anti-inflammatory drugs, physical therapy, and intra-articular injections such as steroid, hyaluronic acid or platelet-rich plasma [4].

Biological therapies in OA are becoming popular. Among various cell therapies, mesenchymal stem cell (MSC) therapy appears to provide encouraging results. Despite, their ability to promote regeneration of chondrocytes and differentiation into cartilage has not been demonstrated in vivo; in response to injury, MSCs are able to release a variety of different molecules, including cytokines and growth factors, with immunomodulatory and trophic effects [5,6]. Hence, several studies reported promising results with the use of autologous microfragmented adipose tissue in the treatment of OA [7,8,9,10]. 

Thus far, MSCs could be isolated from fetal annexes, such as cord blood, placenta, and amniotic fluid [11]. They represent an attractive cell source for allogeneic applications because their MSCs express low levels of MHC class I, no MHC class II and do not induce the activation of allogeneic lymphocyte [12].

Amniotic tissues have been clinically used for many years. Indeed, several studies reported promising results for the treatment of burns [13], skin ulceration and corneal ulcers [14]. More recently, the use of amniotic tissue in orthopedic disorders such as plantar fasciitis, tendinopathies, and cartilage defects is growing [11]. From several preclinical studies, it has been reported that amniotic tissue contains anti-inflammatory cytokines such as IL-10 and IL-1 receptor antagonists as well as inhibitors of metalloproteinases. Moreover, they contain a high content of hyaluronic acid and proteoglycans suggesting a plausible role in OA treatment [15,16]. 

The scope of the present research was to assess the safety, efficacy, and feasibility of intra-articular injections of human amniotic suspension allograft (HASA) in unilateral knee OA in order to evaluate the improvement of clinical status and delay any invasive surgical procedures.

## 2. Materials and Methods

The present pilot study was conducted with the highest respect for individual participants. The procedures followed were in accordance with the ethical standards of the responsible committee on human experimentation (institutional and national) and with the revision of the Declaration of Helsinki, 2014. The Ethics Committee of Verona and Rovigo, Italy approved the study (protocol n. 61,386–19/09/2018 update on 06/02/2019). Inclusion and exclusion criteria were reported in Table 1. Standard weight-bearing anteroposterior, lateral and axial views of the knee were performed to determine the grade of osteoarthritis by Kellgren-Lawrence (KL) classification (grade 1, doubtful narrowing of joint space and possible osteophytic lipping; grade 2, definite osteophytes and possible narrowing of joint space; grade 3, moderate multiple osteophytes, definite narrowing of joint space; and grade 4, large osteophytes, marked narrowing of joint space, severe sclerosis, and definite deformity of bone contour) [17]. Magnetic Resonance Imaging (MRI) of the knee was performed to exclude any other causes of pain (i.e., meniscal flap) that could be addressed surgically. Each patient included in the study underwent all previous conservative treatments (antiinflammatory, physical therapy, intra-articular steroid, viscosupplementation and platelet-rich plasma) without any clinical improvement. 

Patients were evaluated before the treatment and prospectively after 3, 6, and 12 months from the injection. International Knee Documentation Committee (IKDC) score and Visual Analogue Scale (VAS) scores were used for clinical evaluations. Adverse events were also recorded. Baseline characteristics (age, sex, weight, height, smoke, employment status, grade of knee OA, previous surgery, and disease duration) were recorded before the injection.

### 2.1. Human Amniotic Suspension Allograft Preparation

The amniotic membrane was provided by “Fondazione Banca dei Tessuti di Treviso Onlus”. The placenta is sourced from donors undergoing caesarean sections and processed shortly after retrieval, according to Italian requirements. The amniotic membrane is carefully detached from the chorion and rinsed with sterile saline solution to remove residual blood. Afterwards, the amniotic membrane is immersed in a cocktail of antibiotics, validated for human tissues, including vancomycin 100 µg/mL (Hospira), meropenem 200 µg/mL (Fresenius Kabi Italia), gentamicin 200 µg/mL (Fisiopharma) overnight at +4 °C [18,19]. Subsequently, the amniotic membrane is transferred in physiological solution and fragmented with a homogenizer (Polytron PT2500). The suspension obtained is divided into aliquots and stored at −80 °C. Microbiological analyses were performed at several stages throughout the process. 

Donor selection was conducted following Italian directives that foresee a strict clinical and social anamnesis as well as serological (HIV, HBV, HCV, HTLV I/II, Toxoplasmosis, CMV, Lue screening) and molecular screening. The skin was sterilely dressed, and the infiltration was performed on suprapatellar pouch with a 14-gauge needle under ultrasound guidance (15-6 MHz linear transducer). All patients received 3 mL of human amniotic suspension allograft and a series of instructions after injection. In case of discomfort during the treatment, cold therapy and rest were suggested for at least 24 h. Subsequently, mild activities and a gradual resumption of normal sport or recreational activities were allowed as tolerated. Any further therapies such as infiltration or surgery were noted during follow-up.

### 2.2. Statistical Analysis

Data are expressed as a mean, standard deviation (SD) and 95% confidence interval (CI). The following characteristics were considered to stratify the analysis: age of the patient and previous knee surgery. In terms of age, patients were stratified in two classes: age equal to or lower than 50 years and age over 50 years. We verified if the variables were normally distributed through Shapiro–Wilk test. We compared the distributions of variables through nonparametric tests because the variables were not normally distributed. Specifically, we used Wilcoxon rank-test for paired test for IKDC score and VAS to compare preoperative and postoperative values. while a Wilcoxon rank-test for independent sample was implemented to analyze the differences between age groups and in relation of previous surgery. *P*-values less than 0.05 were interpreted as indicative of statistically significant differences. Statistical analyses were performed with statistical software SPSS (version 17.0).

## 3. Results

From January 2020 to January 2021, 30 patients were assessed for eligibility. After inclusion and exclusion criteria, the baseline characteristics of the 25 enrolled patients are summarized in Table 2. There were 11 males and 14 females, with an average age of 45.09 years and a mean Body Mass Index (BMI) of 22.86 kg/m^2^. Knee pain persisted despite prior intra-articular injection of steroid, hyaluronic acid, and Platelet-Rich Plasma in a period of three months to seven years.

The Wilcoxon rank-test was performed for IKDC and EQ VAS scores to find statistically significant differences (*p* < 0.05 for one-tailed tests) comparing pretreatment with after 3 months, pretreatment with after 6 months, pretreatment with after 12 months, 3 months with 6 months results, and 6 months with 12 months results.

The results showed that HASA was statistically effective compared to baseline (Table 3). In fact, for IKDC scores, the results were significant at *p* < 0.0001 after 3 months (z = −2.65), 6 months (z = −4.01) and 12 months t = −3.18).

Similarly, VAS values, the related *t*-test on the data, after 3 months (*z* = −3.52), after 6 months (*z* = −3.82) and after 12 months (*z* = −3.41), showed significant results compared to baseline values (Table 3). 

After 6 months, IKDC and VAS values showed a progressive worsening of the symptoms, even though the differences between 6- and 12 months did not reach significance (Table 3).

Moreover, we stratified the results in relation to age of patients and for previous surgery on the affected knee. We reported the results in Table 4 and Table 5. From analysis we reported that younger patients (<50 years old) were characterized by better clinical results at baseline (Table 4 and Figure 1). However, at longer follow-up the differences were not significant. On the other hand, no clinical differences were reported between patient that underwent previous knee surgery or not (Table 5 and Figure 2).

No severe complications related to the infiltrations were observed during the treatment and the follow-up period. Only minor side effects were detected in four patients (16%), such as transitory intra-articular burning sensation immediately after the injection or mild articular pain for a few days. No patients included in the study underwent any other treatments during follow-up.

## 4. Discussion

The most important finding of the present research is that human amniotic suspension allograft is safe and efficacy for the treatment of pain and improves knee function in OA. As regards to safety, we observed no severe complications related to the injections during the treatment and follow-up period, but only minor side effects that were also common to other infiltration therapies. To the best of our knowledge, only a few clinical studies have been published on the topic [20,21,22,23]. Vines et al. conducted an open-label, single-arm pilot study reporting the safety of amniotic allograft in the treatment of knee OA. However, the study was not powered to assess efficacy including only six patients [20]. Castellanos and Tighe published the results of a prospective pilot study where they reported a significant clinical improvement of symptomatic knee OA. However, they used cryopreserved human umbilical cord and amniotic membrane, and the follow-up was limited to six months [23]. Farr and Gomoll reported the results of a multicenter randomized clinical trial where the administration of amniotic allograft results in a greater clinical improvement compared to hyaluronic acid and saline over 12 months of follow-up [21,22]. 

In light of our results, and in accordance with the previous studies, we reported a significant clinical improvement in pain and function in moderate knee OA. However, by 6 months, IKDC and VAS scores regressed, suggesting that the treatment might not have a lasting effect. However, it is worth mentioning that at 12 months of follow-up, VAS and IKDC scores were significantly better than baseline.

We stratified the patients in the results section according to age and if they underwent previous surgery on the affected knee. Regarding age, we reported that younger patients (<50 years old) were characterized by better clinical results at baseline and 3 months. However, at longer follow-up the differences were not significant, suggesting a slower improvement in the aged group. On the other hand, not clinical differences were reported between patient that underwent surgery or not.

The present study has limitations. The limited size of the sample and the lack of a control group preclude any recommendation about its routine use. The MRI Osteoarthritis Knee Score (MOAKS score) which is a valuable method to stratify patients has not been assessed. However, the present research represents a pilot study, therefore it has not been designed to make a comparison between different grades of OA [24].

It is mandatory to take into consideration that HASA may potentially provoke an immune reaction in the host. However, major histocompatibility antigens (human leukocyte antigen (HLA) classes A, B, C, DR) and β_2_ microglobulin cannot be detected on freshly collected or in vitro cultured amniotic epithelial cells [25]. Moreover, Akle et al. demonstrated that after the subcutaneous transplantation of amniotic membrane in seven healthy individuals, no signs of acute rejection of anti-HLA response were observed [26]. 

However, recent studies pointed out that the administration of high doses of MSCs, as well as other allograft matrices, might induce a humoral response. On the basis of the current knowledge, this response does not appear to promote inflammation or other clinical issues [20,27,28]. Indeed, in a recent trial of HASA injection for knee OA, the level of immunoglobulin (IgA, IgE, IgG, and IgM) levels was tested at baseline, 6 weeks, and 6 months. The study demonstrates the absence of immunologic or adverse reaction to amniotic suspension injection with regards to immunoglobulin or anti-HLA levels [22]. For these reasons, the serum levels of IgG and IgE of the patients have not been tested in the present study.

## 5. Conclusions

Overall, this open-label pilot study indicates that a single intra-articular injection of HASA is safe and efficacy, potentially offering a new minimally invasive therapeutic option for patients who are not eligible for surgeries. At the same time, the physician needs to consider that injection of HASA allograft may potentially provoke immune reactions in the hosts; little is known about the long-term effects of such procedures. Even though, thus far, the procedure is safe. The present allogeneic source obviates the need for additional procedures such as the harvesting of adipose tissue, making this approach at least less invasive [26]. Further high-quality studies are needed to confirm these findings and to compare to other biological solutions.

## Figures and Tables

**Figure 1 jcm-11-03295-f001:**
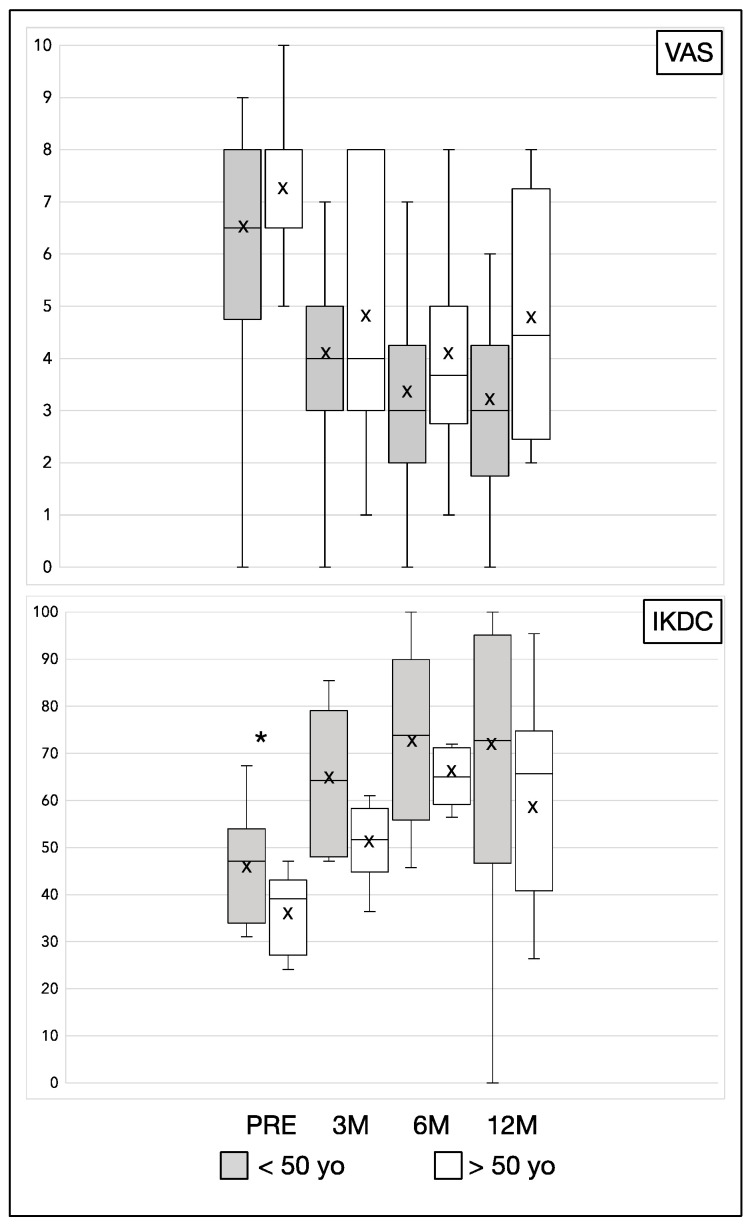
Schematic representation of Table 4 results. PRE: pretreatment values; M: month; yo: years old; ×: mean; *: significant difference (*p* < 0.05).

**Figure 2 jcm-11-03295-f002:**
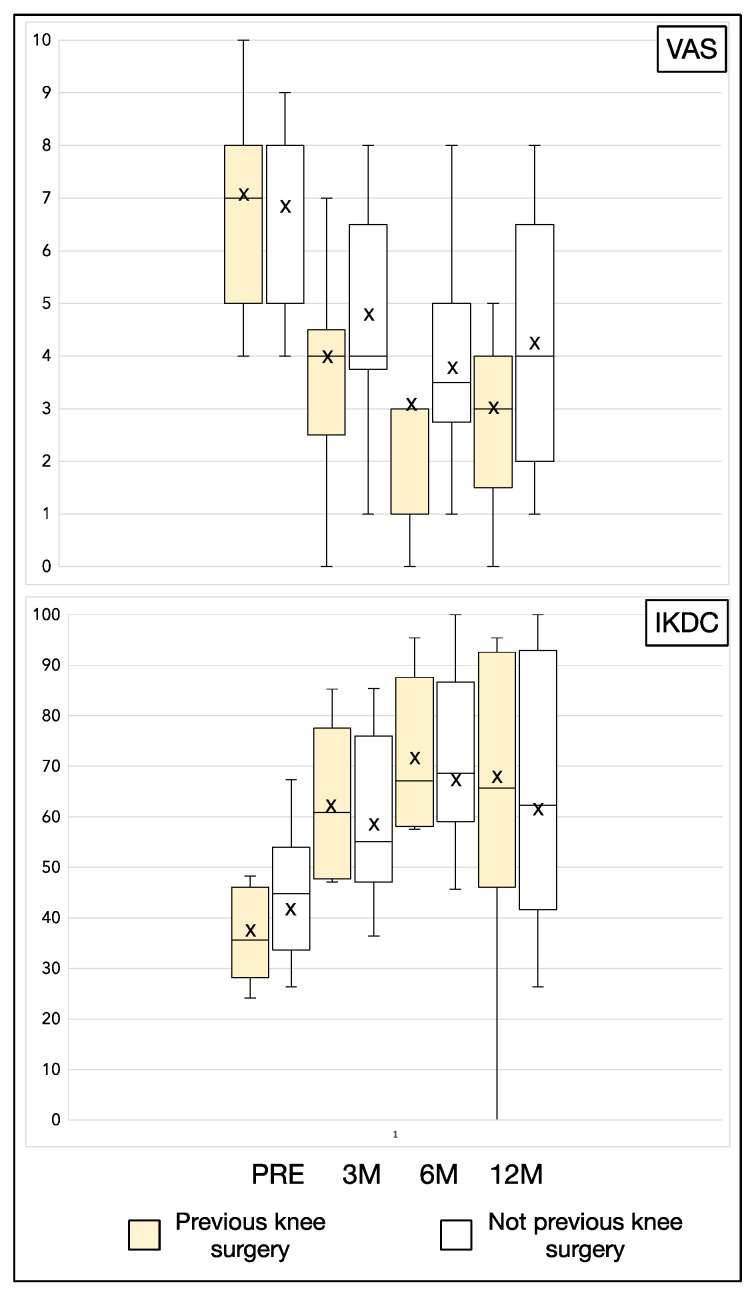
Schematic representation of Table 5 results. PRE: pretreatment values; M: month; yo: years old; ×: mean.

**Table 1 jcm-11-03295-t001:** Inclusion and exclusion criteria. KL: Kellgren-Lawrence; MRI: Magnetic resonance imaging.

*Inclusion Criteria*	*Exclusion Criteria*
Age > 18 years old	Systemic cardiovascular and coagulation disorders or anticoagulant therapy
Degenerative joint changes (KL 1–3)	Degenerative joint changes (KL 4)/surgical MRI findings
Failure of previous conservative treatment (antiinflammatory, physical therapy, intra-articular steroid, viscosupplementation and platelet-rich plasma)	Intra-articular steroid or viscosupplementation injections performed within the last three months.
History of chronic (≥4 months) pain or knee swelling with limitation of daily activities	Rheumatic diseases and septic knee arthritis

**Table 2 jcm-11-03295-t002:** Patient’s characteristics. OA: osteoarthritis; F: female; M: male; SD: standard deviation.

Characteristics	Knee OA (N = 25)
Age (years) [Range]	45.09 (15.31) [22–76]
Sex (F/M)	14/11
Body-Mass-Index (BMI), mean (SD) [range]	22.86 (3.71) [17.9–36.5]
Employment status (worker/retiree)	18/7
Smoker, ex-smoker or non-smoker	5/16/4
Previous surgery on affected knee (Meniscus surgery, osteotomy, cartilage procedure, arthroscopic debridement)	9/16
Radiographic stage (Kellgren Lawrence)	
Grade I	5
Grade II	15
Grade III	5
Disease duration, years (SD)	4.4 (2.3)

**Table 3 jcm-11-03295-t003:** Clinical results. SD standard deviation. PRE: preoperative values; MO: months of follow-up. In bold are indicated significant differences.

Variable	Follow Up	Values		z Value	*p*
IKDC Mean (SD)	PRE	42.47 (11.44)	**PRE versus 3 MO**	−2.65	**0.008**
	3 MO	61.07 (15.3)	**PRE versus 6 MO**	−4.01	**0.000**
	6 MO	71.97 (16.85)	**PRE versus 12 MO**	−3.18	**0.001**
	12 MO	66.80 (25.43)	**3 MO versus 6 MO**	−3.47	**0.000**
			6 MO versus 12 MO	−1.68	0.93
VAS Mean (SD)	PRE	6.95 (1.70)	**PRE versus 3 MO**	−3.52	**0** **.000**
	3 MO	4.45 (1.95)	**PRE versus 6 MO**	−3.82	**0.000**
	6 MO	3.67 (1.74)	**PRE versus 12 MO**	−3.41	**0.000**
	12 MO	3.90 (2.19)	3 MO versus 6 MO	−2.70	**0.007**
			6 MO versus 12 MO	−0.53	0.60

**Table 4 jcm-11-03295-t004:** Clinical results of patients stratified for age. SD: standard deviation; MO: months of follow-up.

Variable	Follow up	≤50 Years Old (N = 13)	>50 Years Old (N = 12)	*p*
IKDC Mean (SD)	PRE	46.76 (11.59)	36.28 (8.31)	**0.03**
	3 MO	66.22 (13.80)	52.7 (14.53)	0.07
	6 MO	74.7 (19.01)	67.54 (12.46)	0.48
	12 MO	72.1 (26.11)	58.2 (23.30)	0.22
VAS Mean (SD)	PRE	6.62 (1.61)	7.44 (1.81)	0.22
	3 MO	4.15 (1.34)	4.89 (2.62)	0.49
	6 MO	3.46 (1.45)	4.00 (2.20)	0.44
	12 MO	3.31 (1.55)	4.88 (2.80)	0.23

**Table 5 jcm-11-03295-t005:** Clinical results of patients stratified for previous knee surgeries. SD: standard deviation; MO: months of follow up.

Variable	Follow up	Previous Knee Surgery (N = 10)	Not Previous Knee Surgery (N = 15)	*p*
IKDC Mean (SD)	PRE	38.83 (8.55)	43.83 (12.82)	0.36
	3 MO	63.87 (16.11)	58.1 (14.68)	0.65
	6 MO	73.54 (16.10)	68.97 (16.33)	0.81
	12 MO	69.53 (23.49)	62.78 (26.24)	0.71
VAS Mean (SD)	PRE	7.13 (1.81)	6.86 (1.70)	0.91
	3 MO	4.00 (1.51)	4.71 (2.16)	0.38
	6 MO	3.29 (1.70)	3.86 (1.79)	0.42
	12 MO	3.00 (1.41)	4.36 (2.41)	0.27

## Data Availability

The data presented in this study are available on request from the corresponding author.

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
