# Peer review of "Human Amniotic Suspension Allograft Improves Pain and Function in Knee Osteoarthritis: A Prospective Not Randomized Clinical Pilot Study"

_jcm, 2022, doi:10.3390/jcm11123295_

Round 1
Reviewer 1 Report
This pilot study evaluated the safety and effectiveness of human amniotic allograft injection into knee joints as a possible OA treatment.
This study was conducted according to generally accepted protocols, however there are some limitations – most of them mentioned by the authors
- I would suggest including “pilot study” in the title to clarify the nature of the study for the readers.
- The authors used Xrays / Kellgren – Lawrence as inclusion criteria; this scale has well described limitations and is not a precise tool to say the least; MRI and MOAKS or other scoring system would be a better choice for this study . Perhaps some patients (KL-1) had problems related to degenerative meniscal lesions ? Please mention this limitation
- The authors should mentioned that the injection of HAC allograft may potentially provoke immune reactions in the hosts; little is known about long-term effects of such procedures; the authors barely mentioned that they did not evaluate Ig levels in their patients
- This “immunological” problem is also my biggest moral concern – how were donors/recipients matched – for instance was the blood type included as a criteria ? Please provide an in-depth description how the risk of patient immunization was minimized in this study
- I can only assume that donors were tested for Hep-B, Hep-C, HIV etc. please confirm this in the materials section
- The conclusion should include a statement on possible immunization as mentioned previously
- This is a llesser point – please consider substituting the word “Anyways” in your discussion – this would improve the quality of your English.
Reviewer 2 Report
The current manuscript entitled “Human amniotic suspension allograft improves pain and function in knee osteoarthritis: a prospective not randomized clinical study" is novel and of scientific interest. However, there are several limitations that need to be addressed.
The current study major limitation is the fact that the control group arm was not included neither the patients were randomized. Furthermore, the fact that prior IA interventions were allowed (after minimum of 3 months) is another cofounding variable for the interpretation of the results. The authors should address these issues and how can this impact the reported results.
The authors stated that “Any further therapies as infiltration or surgery were noted during follow-up.” The authors show point how many of those patients proceed to other further therapies, and analyze the results in light of this data.
It would be interesting to know the impact of HASA per OA KL groups.
The current article needs extensive statistical revision. The patient groups is very heterogenous as the authors highlighted, hence normal distribution cannot be assumed (was also not tested). The authors should consider non-parametric tests.
Round 2
Reviewer 1 Report
thank you for the reviewed version